# A Brief Report on the Collapse of Self-Built Houses on 29 April 2022, in Changsha, China

**DOI:** 10.3390/ijerph20010061

**Published:** 2022-12-21

**Authors:** Yan-Ning Wang, Qian Chen, Jing-Rui Peng, Jun Chen

**Affiliations:** 1Department of Civil and Environmental Engineering, Shantou University, Shantou 515063, China; 2State Key Laboratory for Geomechanics and Deep Underground Engineering, China University of Mining and Technology, Xuzhou 221000, China; 3Department of Civil Engineering, Shanghai Jiao Tong University, Shanghai 200030, China

**Keywords:** house collapse, bearing capacity of foundation, engineering accident, design guidelines

## Abstract

A self-built house collapsed in Changsha City, Hunan Province, on 29 April 2022, causing 53 people to be trapped and 39 people to go missing. The tragedy caused a huge loss of life, so the stability of self-built houses has a non-negligible impaction on public safety. This report presents causes of the incident, rescue operations, cause analysis, and the analytic hierarchy process (AHP). The main conclusions about the reasons for the collapse include: (a) reckless heightening, (b) unreasonable structure, (c) adverse geological conditions, (d) architectural vulnerabilities, (e) negligence of competent authorities, and (f) lack of security awareness. This paper provides a reference for avoiding similar security incidents in the future.

## 1. Introduction

The current status of self-built house construction and housing safety are essential elements of new rural construction and are related to the pace of urbanization. Along with the development, a series of construction defects have been revealed. Design guidelines suggest that the life span of a building should be between 50 and 100 years. However, this building collapsed after only 20 years [1]. This accident was caused by unscientific site selection, poor design, and unprofessional construction approaches. In addition, the uneven scale of self-built houses and the lack of centralized living add to the difficulty of management and put pressure on the quality and safety management of self-built houses. Although technical specifications and guidelines are available for industry development, many problems, such as unqualified housing construction planning and uncontrolled construction quality, still emerge in construction projects. These problems have led to a series of construction disasters. This paper discusses a serious building failure disaster in Changsha on 29 April 2022 which caused the death of 53 people and left 39 people missing. If a self-built house is established near a college or residential block, more tenants and homeowners will quasi-seek more significant benefits. This may cause a severe public safety incident with a significant adverse impact on society [2,3,4]. 

There are different views on building collapse accidents in China and abroad (see Figure 1). From Ayodeji and Adedeji’s point of view, building collapse is a state of total failure where the structure has given way and most elements have collapsed or are about to collapse [5]. Fagbenle and Olawunmi consider the collapse of a building as the inability of the building to meet the expected load-bearing structure [6]. In addition, Oragonju et al. argue that building collapse is the total or partial failure of one or more constituent structures that prevent the building from performing its stability, safety, and comfort roles [7]. In a series of building collapses, Daniel et al. concluded that a large part of the building collapse was due to some part of the column being unstable, causing the panel on the column to lose support and thus triggering the collapse [8,9]. Byfield et al. argued that the lack of alternative paths capable of redistributing the load after column failure leads to building collapse [10,11]. In conclusion, building collapse may affect the whole building or some parts of the building. Safety and stability are the most critical issues in this regard [12,13]. This study analyzes building collapses in China and applies risk assessment methods to analyze critical factors. This can be used to compare building collapse situations in China and abroad, summarize the similarities and differences, and make targeted observations to prevent the recurrence of similar cases of such building collapse in China and abroad [10,14]. In our country, the research results on analyzing the causes of building collapse accidents have been very mature. Some scholars have studied the characteristics of the spatial and temporal distribution of collapse accidents in China from 2000 to 2020. These studies identify the high incidence period, high incidence area, and accident causes, and summarize the different characteristics of collapse accidents [15]. 

In recent years, the construction industry in China has proliferated and the number of construction projects has increased, resulting in building collapse accidents that are prone to occur. Most of these accidents could have been avoided, but similar cases remain. Therefore, we call on the relevant departments to strengthen the safety supervision of housing construction and promote the overall renovation of self-built houses to solve the housing safety problem [16,17]. This study analyzes such accidents and makes recommendations of great significance and practical value in preventing the recurrence of similar incidents. 

## 2. Materials and Methods

### 2.1. Overview of Exploration Event

A resident’s self-built house collapsed in Panshuwan, a street in Jinshanqiao, Wangcheng District, Changsha City. It is located in the northern part of Xiangjiang New Area, with an area of 24.07 square kilometers, which was divided from the original Jinshan Street in August 2012 and is adjacent to Wangcheng District. It is 8 km from the municipal party committee and the municipal government’s residence and is a double core area of Xiangjiang New Area and Wangcheng Economic Development Zone. It is the main component street of the new core area of the high-speed railway west city created by Changsha. There are seven communities under its jurisdiction, with a resident population of 20,858. The collapsed house is located in Panshuwan, Jinping Community, Jinshanqiao Street, as shown in Figure 2 and Figure 3 [18].

At around noon on 29 April 2022, this self-built house in Wangcheng District collapsed instantly, raising a sky full of dust. After the building collapsed, the walls’ debris piled up two stories high. According to a report on 6 May, the search and rescue work at the site of the “4.29” particularly significant resident self-built house collapse in Changsha City has been completed, with a total of 10 people rescued and 53 dead at the accident site [18]. 

### 2.2. Rescue Effort

After the accident, the relevant departments quickly started the emergency plan and set up the scene dispatch and disposal command center. They urgently performed on-site rescue, with an emergency dispatch of more than 700 fire rescue personnel, armed police, social emergency forces officers, and soldiers, as well as mobilizing fire trucks, large cranes, power generators, MIMO radar life detectors, and other rescue equipment totaling more than 120 units, to undertake an on-site search and rescue. The relevant progress is shown in Figure 4a,b.

The collapsed house squeezed the houses on both sides, to the extent the neighboring houses were seriously damaged and may collapse at any time. In order to prevent the occurrence of secondary disasters, the expert group and rescue team on-site meeting is actively searching and rescuing. They conducted emergency reinforcement of self-built houses and immediately transferred the surrounding people. They also did an excellent job of safely identifying the houses around the site to eliminate safety hazards. This fully protects the site environment for construction and rescue and smooths the passage for large rescue materials and casualty rescue [19]. 

In addition, according to the relevant reporters, the relevant departments were trying to perform personnel rescue and treatment work. They coordinated authoritative experts, deployed high-quality medical resources, rushed to the scene to guide the treatment, opened green channels, transferred the injured, and did their best to reduce casualties. The five injured people were sent to the hospital for treatment, and vital signs were stable. The public security department continued to increase the technical and big data investigation to further verify the number of trapped and missing persons and their specific circumstances. At the same time, the relevant departments arranged for exceptional staff to perform an excellent job of caring for family members and comforting injured people, providing emotional counseling work, and organizing related aftercare, as shown in Figure 4c,d. 

## 3. Analysis and Discussion

### 3.1. Cause Analysis and Reflections

#### 3.1.1. Recklessly Heightening

Similar to many previous collapses, photos of the building before it collapsed can be quickly found online to analyze the original structure of the building. As shown in Figure 5, the building appears to have been topped with a two-story attic by the owner, resulting in a total of eight (or seven and a half) stories. Figure 6 illustrates how the owner’s self-built house has become taller over the past decade. This self-built roof covering is not an isolated phenomenon. The covered floors are colored steel tile roofs. This material is easy to construct and inexpensive. Even if violations are found and need to be removed, it causes little economic damage compared to reinforced concrete and is used by many homeowners. However, while covering, it can also bring significant safety risks, quickly leading to uneven settlement of the foundations, and collapse. Similar incidents had occurred in many places before the collapse of self-built houses in Changsha. [20]. 

#### 3.1.2. Internal Structure

By analyzing the internal structure of each floor from the bottom up, it is known that the building has corresponding structural hazards. Since the corresponding internal structural layout was not found from the bottom up, the style of similar buildings in the surrounding area was analyzed to infer the structural characteristics of the original building. As seen in Figure 7a, no apparent beams and columns were found in the internal structure of the building on the 4–6th floors, which are all brick and concrete structures. However, when we walk to the dining room on the 2nd floor, we can find the corresponding transverse structural beams on the top but no longitudinal structural beams. This leads to the transfer of forces from the transverse load-bearing beams directly through the brick masonry to the floor, and this collapse occurs when the load is high, as can be seen in Figure 7b [21].

#### 3.1.3. Geographic Information Condition

Changsha Wangcheng District has medium geological and environmental conditions which are mainly slate, mudstone, sandstone, conglomerate, shale, and granite, with high lithology. Its resistance to weathering is poor. The buried depth of groundwater is generally 15 m ≤ 30~50 m, with poor water richness, such as bedrock fracture water and abundant karst water, without adverse geological effects. Changsha has a subtropical monsoonal humid climate. The temperature rises significantly from late April throughout the year and the average monthly temperature can exceed 25 °C. At the same time, rainfall also increases significantly, and Changsha has entered the rainy season with a monthly average rainfall greater than 150 mm (see Figure 8). However, the incident occurred in a rainy area in the south–central part of Wangcheng District. The increase in rainfall resulted in continuous softening of the foundation soil, increased pore water pressure, increased compressibility, and a significant decrease in shear strength and bearing capacity. The uneven loading caused significant differential settlement of the foundation, exceeding the requirements of the relevant codes and subsequent collapse. 

Human engineering activities (mining, road construction, etc.) are flourishing, so geological hazards are commonly developed. There are many geological hazards accompanying human engineering activities. According to the statistics of geological hazard survey results in Wangcheng District, Changsha City, in 2008 (see Table 1), a number of relevant and geological hazard points were surveyed [22]. 

According to relevant information, geological hazards in Wangcheng District, Changsha City, have a specific regional nature in distribution according to different types, while the development also has specific regional differences. Except for the northern plain, all 11 townships in Wangcheng have geological disaster points distributed with different degrees of hazards, with those townships having more disaster points being Chating Town, Tongguan Town, Dingzi Town, Gaotangling Town, etc. The geohazard sites are mainly distributed in the low mountain and hilly areas, near the settlements living by the ditches and the sides of the rural roads under construction. Jinshan Street is located in the southern part of Wangcheng District. The fracture zone in the southern area provides material sources for disasters such as landslides. At the rock level, the joint fracture surface often becomes the controlling structural surface for collapses or landslides. Hence, the southern part of Wangcheng District belongs to the geological disaster-prone area. Where the self-built house collapsed, the indoor and outdoor floors were hardened and closed, making further inspection impossible [23,24]. 

#### 3.1.4. Analytic Hierarchy Process (AHP)

Hierarchical analysis (AHP) is a standard accident analysis method combining qualitative and quantitative methods to calculate the weight of each factor. In this report, the factors of accident causes were extracted from the accident investigation report and summarized from four aspects: human factor, material factor, management factor, and environmental factor. The factors summarized are: Detection unit fraud, Insufficient security awareness, Inadequate structure, Structural heightening, Inadequate regulation, Inadequate safety education, Continuous rainfall, and Inadequate geological conditions. In this report, 10 experts—including scholars, field engineers, and designers—were invited to score the relevant factors. The weights of each factor can be calculated by combining different experts’ scores (see Table 2) with yaahp software, as shown in Figure 9a. Among them, the judgment matrix’s consistency (CR) test is required. These data’s consistency index (CR) is 0.0454 < 0.1, so they meets the consistency requirement. Based on the hierarchical analysis (AHP) results, a fuzzy comprehensive evaluation analysis was conducted and the degree of importance of different factors was obtained, as shown in Figure 9a. 

Sensitivity analysis can determine the impact that a change in a factor’s weight has on the overall weight. This report focuses on sensitivity analysis through V12.9 version of Spssau software. The degree to which the uncertainty of each element affects the target is examined by holding all other factors at their baseline value [25]. The analysis reveals that Insufficient security awareness and Inadequate structure have the most significant impact on the weights (see Figure 9b). This shows that the related analysis method learns that people’s awareness and building structure have the most critical influence. This provides an essential direction for the development of future preventive measures [26,27,28]. 

### 3.2. Main Factors of Explosion

(1) **Poor quality of housing construction:** A good building design must have strength characteristics that can withstand the structure’s expected dead and live loads. Failure to meet this essential requirement usually overloads the building, which can lead to its collapse. In order to solve the problems of the general population, it is known that many places build fast food-style buildings that need to meet the load expectations. This will lay a safety hazard for the building collapses that have occurred in recent years [30].

(2) **Non-compliance with standards:** The quantity and quality of the specified building materials must be sufficient for the building’s needs. Quality materials are critical to making the building durable, as specified in the standard specifications. The use of substandard quality materials in construction activities is a root cause of building collapse and failure. Last century, self-built houses were prone to collapse, while some engineering firms also cut corners and built “tofu-dreg” buildings for profit, for example, the collapse of self-built houses in Changsha. The engineering unit concerned omitted even vertical load-bearing beams to control costs and carried the load through brick masonry, which seriously affected the quality and service life of the house, as evidenced by the collapse of some buildings in recent years.

(3) **Abuse of building use:** In many urban villages and rural areas, some homeowners add layers privately and renovate savagely to collect rent or open a store, laying down hidden dangers for future house collapse. In the case of the self-built house collapse in Changsha, this was one of the direct causes of the accident. For renovation and extension programs, the load-bearing capacity and structural integrity of the building must be checked before and after the removal of unsuitable programs, which must be performed by experts (structural engineers) [31].

## 4. Recommendations

According to the study, the collapse of self-built houses in Changsha is primarily attributed to human error. It is usually accompanied by varying damage, including human life and property. While building collapse cannot be eliminated, compliance, satisfactory performance, standards, and construction quality can be ensured through the cooperative and collaborative role of architects, relevant regulatory bodies in the construction industry, related professions, and other relevant stakeholders in the industry [32]. 

(1) For the corresponding structural repair, reinforcement, structural correction, and reduction, the relevant departments should apply appropriate methods to reinforce and repair the damaged part of the substructure promptly in strict accordance with the relevant design specifications and restore the regular use function of the building to achieve the purpose of safe load-bearing. 

(2) Rigorous treatment of wall surfaces or component surfaces can leave large- and small-size cracks. Treating these cracks is necessary both to beautify the appearance of the building and to not leave hidden problems, thus further aggravating the wall cracks of the composite housing structure. 

(3) The government should establish a supervisory force to ensure that the relevant authorities approve each building design and that construction and approval are performed through relevant regulations. The viewpoint of each construction stage should be evaluated through regular visits to construction sites. These evaluations should mainly: a. Strengthen the management of each design aspect of construction projects; b. Strengthen the review and evaluation of construction enterprise qualifications; c. Strengthen the management of construction materials; d. Accelerate the establishment and improvement of construction quality management system; e. Increase the supervision of quality management [33,34]. 

(4) Building collapse accidents are characterized by high suddenness, high destructive power, and high social impact. Therefore, the relevant departments should play a key role in emergency treatment (see Figure 10). Thus, the rescue system for building collapse accidents can be continuously improved to enhance the rescue capability [35]. 

## 5. Conclusions

This study discusses several reasons for the collapse of self-built houses in Wangcheng District, Changsha City, on 29 April 2022. The main conclusions are as follows: 

(1) The collapse of self-built houses in Changsha was caused by the design not conforming to the specifications. Therefore, it can be seen that the relevant units’ compliance with the norms and the control of quality need to be put in place, and the supervision of the house should be tightened to prevent similar incidents. 

(2) The emergency departments should respond actively, put life first, use the golden time of rescue, be responsible for the safety of the victims, and minimize casualties. 

(3) Lack of safety awareness among people and the height of houses are among the most critical causes of house collapses. People’s construction concept should be strengthened to minimize the occurrence of such events. On the other hand, the education of managers and the study and application of building codes and standards by engineers and technicians should be strengthened [37]. 

## Figures and Tables

**Figure 1 ijerph-20-00061-f001:**
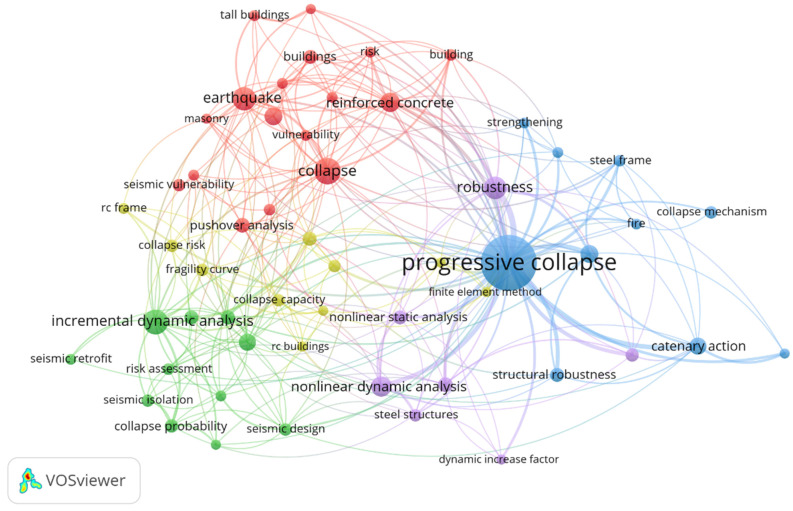
High-frequency keyword map in the field of building collapse.

**Figure 2 ijerph-20-00061-f002:**
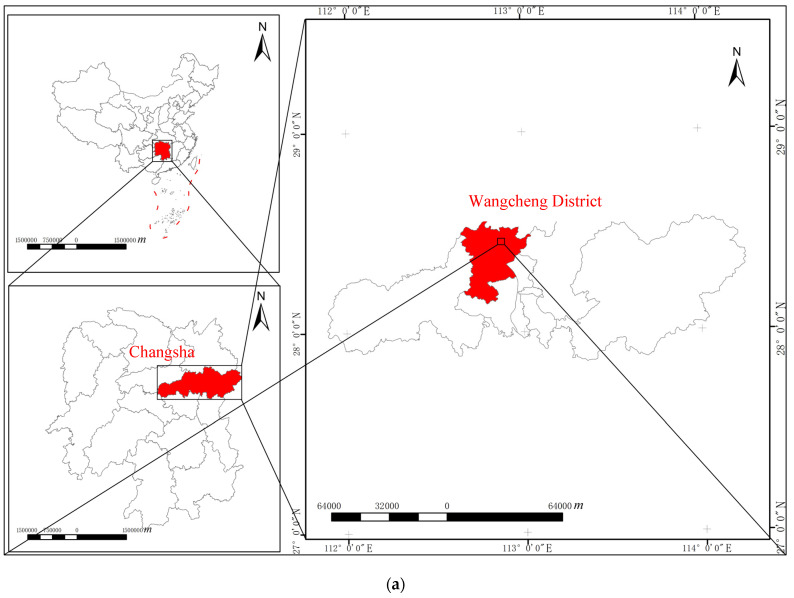
Location of the accident occurrence area: (**a**) Location map and image; (**b**) Accident satellite map.

**Figure 3 ijerph-20-00061-f003:**
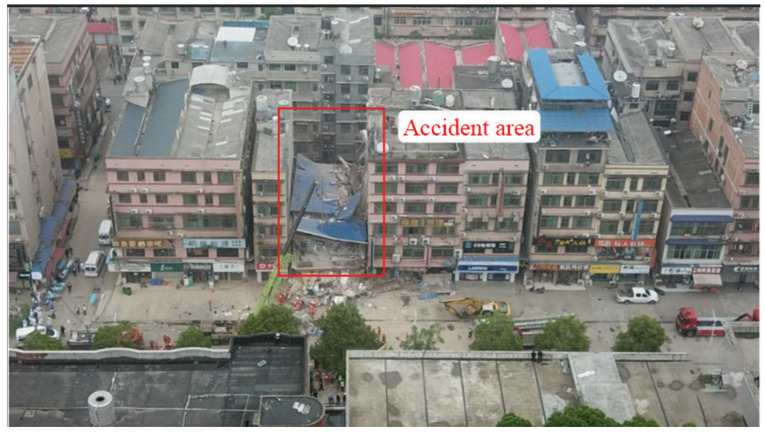
Photo of the collapse site.

**Figure 4 ijerph-20-00061-f004:**
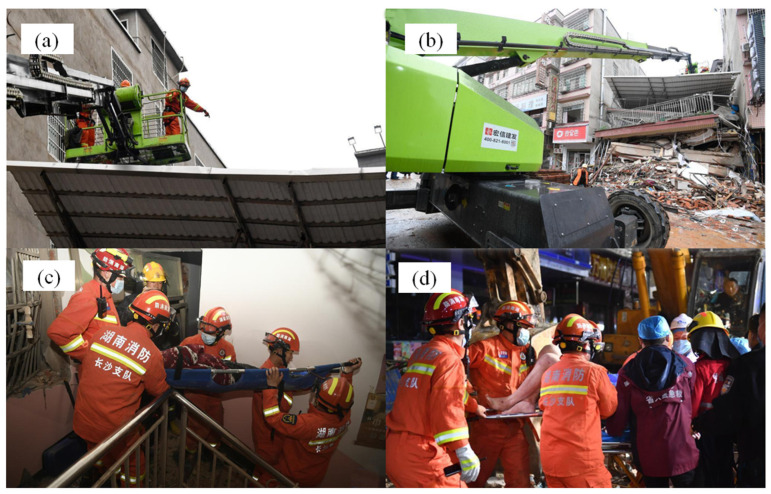
(**a**,**b**) Rescuers perform rescue operations; (**c**,**d**) Rescue work by relevant personnel.

**Figure 5 ijerph-20-00061-f005:**
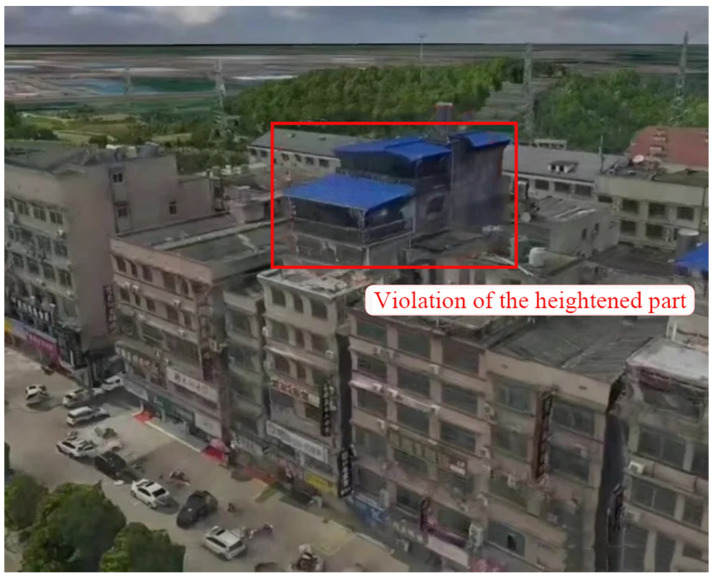
Collapsed building original structure.

**Figure 6 ijerph-20-00061-f006:**
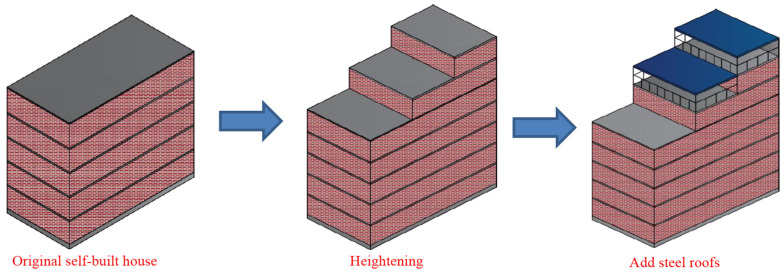
Remodeling of the self-built house over the past decade.

**Figure 7 ijerph-20-00061-f007:**
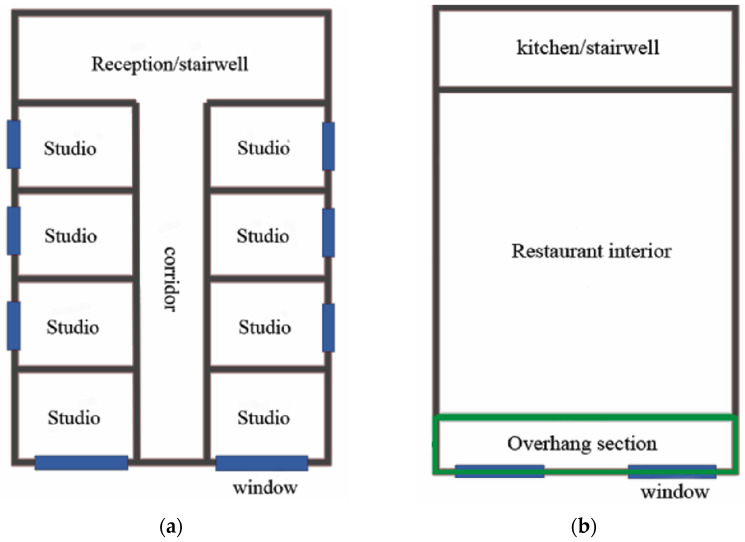
(**a**) Internal structure diagram of layers 4–6; (**b**) The internal structure of the restaurant on the 2nd floor.

**Figure 8 ijerph-20-00061-f008:**
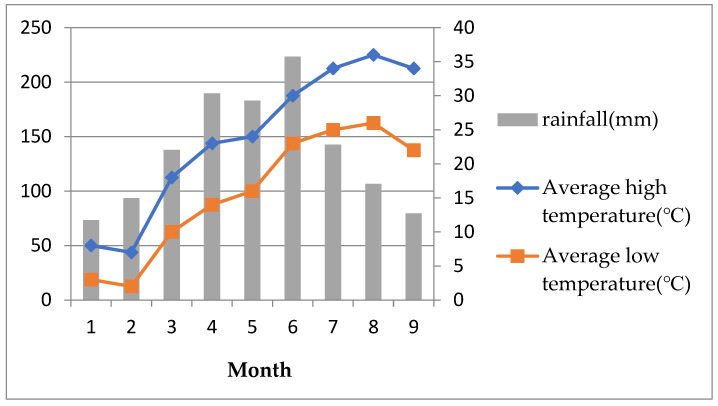
Rainfall, average high temperature, and average low temperature in Changsha.

**Figure 9 ijerph-20-00061-f009:**
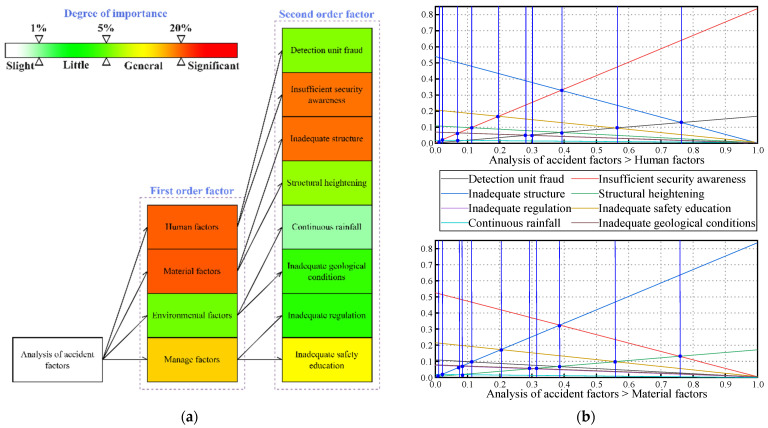
(**a**) The analytic hierarchy process method derives the weight of each influencing factor; (**b**) Analysis of sensitivity for human factors and material factors [29].

**Figure 10 ijerph-20-00061-f010:**
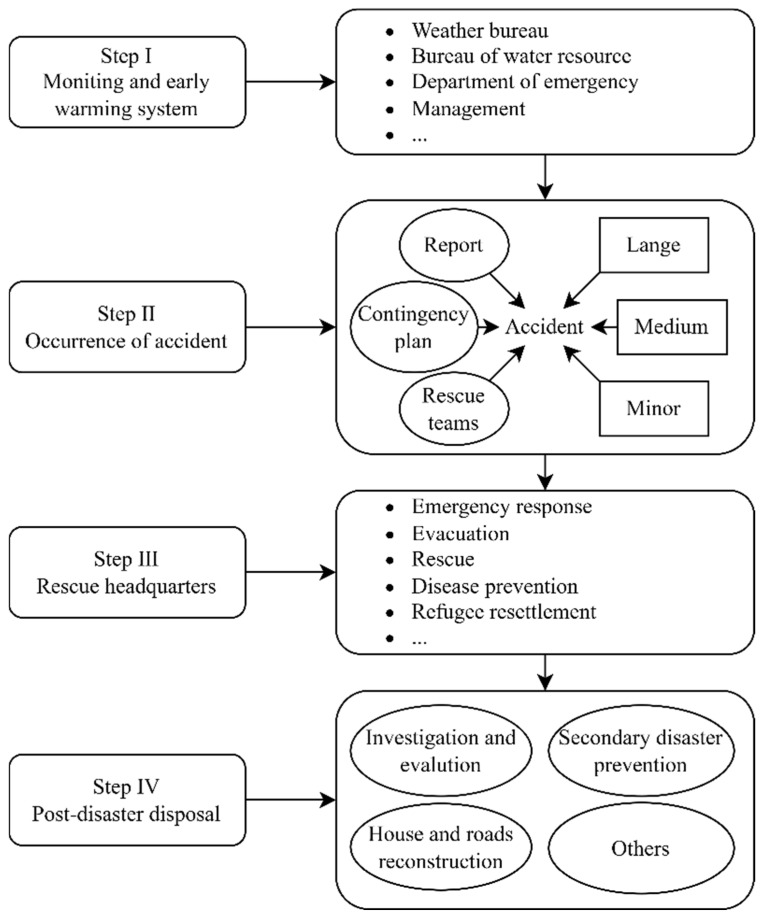
Flow chart of geological disaster emergency treatment in China (recreated from Yantao [36]).

**Table 1 ijerph-20-00061-t001:** List of geological disasters in Wangcheng District, Changsha City.

Investigation Area (km^2^)		Investigate Townships		Investigate Administrative Villages and Neighborhood Committees			
951.1		15		156			
Geological disaster point	total	landslide	collapse	debris flow	ground fissures	ground collapse	unstable slope
	49	14	15	1	0	4	15
	death toll		0	direct loss (CNY 10,000)		532	
Geological hazards	total	potential landslide	potential collapse	potential debris flow	unstable slope	potential ground collapse	others
	34	7	7	1	15	4	Plumbing 13 places
	Threat population		619	Threatened property (CNY 10,000)		12,430.5	

**Table 2 ijerph-20-00061-t002:** (**a**) Score experts on decision-making factors; (**b**–**e**) Score experts on intermediate factors.

(**a**)				
	**Human Factors**	**Material Factors**	**Management Factors**	**Environmental Factors**
Human factors	1.0	1.0	3.0	5.0
Material factors	1.0	1.0	3.0	7.0
Management factors	0.3	0.3	1.0	5.0
Environmental factors	0.2	0.1	0.2	1.0
(**b**)				
	**Detection Unit Fraud**		**Insufficient Security Awareness**	
Detection unit fraud	1.0		0.2	
Insufficient security awareness	5.0		1.0	
(**c**)				
	**Inadequate Structure**		**Structural Heightening**	
Inadequate structure	1.0		5.0	
Structural heightening	0.2		1.0	
(**d**)				
	**Inadequate Regulation**		**Inadequate Safety Education**	
Inadequate regulation	1.0		0.3	
Inadequate safety education	3.0		1.0	
(**e**)				
	**Continuous Rainfall**		**Inadequate Geological Conditions**	
Continuous rainfall	1.0		0.2	
Inadequate geological conditions	5.0		1.0	

## Data Availability

The datasets generated and/or analyzed during the current study are available from the corresponding author on reasonable request.

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
