# Peer review of "A Brief Report on the Collapse of Self-Built Houses on 29 April 2022, in Changsha, China"

_ijerph, 2022, doi:10.3390/ijerph20010061_

Round 1

Reviewer 1 Report

Your research and the report are generally good. I could recommend the three changes to make your work more attractive to readers. First, this report fails to clearly state the research significance and conclusions in abstract. Second, the Section 3.1.4 should be elaborated in more detail to enhance the credibility of conclusions, including the number of experts, the tool for sensitivity analysis, and the interpretation of Figure 9. Third, the subtitle of the report is suggested to be adjusted. Section 2 seems to be the background of the accident, and Section 3 is the research method. The results and recommendation are suggested to be described in a separate section.

Author Response

We apologize for the possible misunderstanding of the editor’s and reviewers’ intention and we are willing to revise it based on further guidance. The comments from the respected editor and reviewers provided great help for our article to better summarize the results and present the research conclusions and significance more clearly and intuitively. So that the revised manuscript can be more convenient for the relevant researchers to reference and inspect and get more valuable information. We hope that the revised manuscript will be judged appropriate for publication in International Journal of Environmental Research and Public Health.

The comments are all valuable and very helpful for revising and improving our paper, as well as the important guiding significance to our research. We appreciate for the Editor and Reviewers’ warm work earnestly and hope that the correction will meet with approval. Once again, thank you very much for your comments and suggestions.

Author Response

(The authors gave the same response as above.)
